# A Deep Learning Approach to Classify Sitting and Sleep History from Raw Accelerometry Data during Simulated Driving

**DOI:** 10.3390/s22176598

**Published:** 2022-09-01

**Authors:** Georgia A. Tuckwell, James A. Keal, Charlotte C. Gupta, Sally A. Ferguson, Jarrad D. Kowlessar, Grace E. Vincent

**Affiliations:** 1School of Health, Medical and Applied Sciences, Central Queensland University, Adelaide 5001, Australia; 2School of Physical Sciences, The University of Adelaide, Adelaide 5005, Australia; 3College of Humanities and Social Sciences, Flinders University, Adelaide 5005, Australia

**Keywords:** machine learning, neural network, prolonged sitting, driving, sleepiness, fatigue

## Abstract

Prolonged sitting and inadequate sleep can impact driving performance. Therefore, objective knowledge of a driver’s recent sitting and sleep history could help reduce safety risks. This study aimed to apply deep learning to raw accelerometry data collected during a simulated driving task to classify recent sitting and sleep history. Participants (n = 84, Mean ± SD age = 23.5 ± 4.8, 49% Female) completed a seven-day laboratory study. Raw accelerometry data were collected from a thigh-worn accelerometer during a 20-min simulated drive (8:10 h and 17:30 h each day). Two convolutional neural networks (CNNs; ResNet-18 and DixonNet) were trained to classify accelerometry data into four classes (sitting or breaking up sitting and 9-h or 5-h sleep). Accuracy was determined using five-fold cross-validation. ResNet-18 produced higher accuracy scores: 88.6 ± 1.3% for activity (compared to 77.2 ± 2.6% from DixonNet) and 88.6 ± 1.1% for sleep history (compared to 75.2 ± 2.6% from DixonNet). Class activation mapping revealed distinct patterns of movement and postural changes between classes. Findings demonstrate the suitability of CNNs in classifying sitting and sleep history using thigh-worn accelerometer data collected during a simulated drive. This approach has implications for the identification of drivers at risk of fatigue-related impairment.

## 1. Introduction

Safety-critical tasks such as driving require a range of skills and competencies that must be performed at a high level [1,2,3]. Psychomotor functions required for driving are known to be impacted by inadequate sleep [4]. Specifically, reduced psychomotor functioning as a result of inadequate sleep manifests in impaired reaction time and precise motor responses [5,6,7]. With specific reference to driving, research has reported delayed response times in drivers with inadequate sleep, with drivers taking up to 44% longer to start braking [8]. Drowsy or fatigued driving represents a significant risk to driver safety, with 20% of fatalities globally reported to be fatigue-related [5,8]. Importantly, impairments in psychomotor functioning, like those seen following inadequate sleep, have also been reported following periods of prolonged sitting [4,9,10,11,12]. Prolonged sitting was associated with longer reaction times when compared to breaking up sitting with physical activity [13]. Given that both inadequate sleep and prolonged sitting can impact driving performance, the objective assessment of sitting and sleep history prior to or in the early stages of a drive may provide an avenue to reduce the risk of road accidents.

Recent developments in deep learning for driver fatigue detection have relied on computer vision (information derived from images) using driver-facing cameras [14]. Recordings of drivers have assessed physiological and behavioural features such as pupil dilation, yawning and blinking [14,15,16,17,18] and demonstrated a high degree of accuracy for driver fatigue detection [17,19,20]. Computer vision approaches rely primarily on facial feature detection, but they provide no information about vehicle control or performance (e.g., speed and lane variability) [21]. An analysis of driver behaviours at the interface with vehicle controls could provide additional insights into the relationship between fatigue and driving performance. 

Driver fatigue detection approaches have also utilised both physiological- and vehicle-based measures [15]. Physiological-based measures for driver fatigue detection include brain activity (i.e., ocular activity, muscle tone, cardiac activity) and respiration rate [22,23,24,25,26]. Current vehicle-based measures for driver fatigue detection include steering wheel angle and lane deviation analysis [19]. One study, which integrated physiological signals (respiratory signals and pulse rate) with vehicle-based data (steering wheel angle), using a support vector machine approach, classified driver drowsiness with an accuracy of up to 96% [27]. While effective, the current physiological driver fatigue detection methods can be expensive and require labour-intensive data processing [7,23]. A single signal that captures the physical movement during driving may be a more cost-effective means of detecting fatigue. The assessment of physical movement may also provide information about how movement changes with fatigue and how such changes impact driving performance. Critically, the assessment of physical movement may also be used to classify sitting history, which impacts aspects of performance relevant to driving but currently has an unknown relationship to driving performance. A low-cost body-worn device, such as an accelerometer, may provide a means to capture a movement signal that can be used as an indicator of driver state.

Body-worn sensors, such as triaxial accelerometers (i.e., three axes: x, y, z), can collect high-resolution physical movement data (e.g., 20 samples per second) [28]. While the physical movements required during driving (steering, braking and accelerating events) can be recorded using such accelerometers [29], detecting subtle and systematic changes in movement requires a computationally efficient methodology that can recognise and extract patterns within large data sets. Previous research has demonstrated the value of a deep learning approach to interrogate raw accelerometry data. Recently, Dixon et al. [30] demonstrated the use of a convolutional neural network (CNN) to detect subtle patterns within accelerometer data. Dixon et al. [30] used data from body-worn accelerometers to classify three different terrain surfaces traversed at running pace (concrete, synthetic and woodchip) with an accuracy of up to 97%. An accelerometer placed on the thigh of a driver will capture leg movements as they relate to the pedal inputs (acceleration and brake) of the vehicle control. This may be sufficient to detect subtle patterns related to sitting and sleep history in drivers using a CNN approach. 

A CNN is a deep learning architecture inspired by the visual cortex of an animal [31]. The networks are designed to automatically extract features in the form of translation-invariant spatial relationships [32], and thus, they excel at pattern detection in large datasets of spatially-dependent variables, such as images [32,33,34]. Automatic feature extraction is an important consideration for datasets in which the pertinent signal features (i.e., changes in measured acceleration indicative of driver state) are unknown. CNNs are used ubiquitously in supervised machine learning to differentiate between previously defined classes in data (i.e., conditions: sitting/breaking up sitting and sleep history) [33]. Further, the post-hoc analysis of a trained CNN model, such as class activation mapping (CAM), can be used to identify and visually present the learned features which have been proven as keys to class differentiation [35,36]. 

The primary aim of this study was to apply deep learning to raw accelerometry data collected during a simulated driving task to classify the recent sitting and sleep history of the driver. A secondary aim was to use the post-hoc analysis of the trained models to characterise differences in physical movement between the two classes (i.e., sitting and sleep history). This analysis may provide the basis for a novel, accessible and more feasible method of detecting at-risk drivers through physical movement relating to vehicle control. 

## 2. Materials and Methods

### 2.1. Study Design

The approach takes four phases: experimental study (data collection), CNN training, CNN evaluation and class activation mapping (CAM). The study was a laboratory-based, randomised, between-subjects, factorial (2 × 2) experimental design. Participants were randomly allocated to one of four conditions, consisting of a sleep opportunity condition of 9-h or 5-h (time in bed) and an activity condition of sitting or breaking up sitting (further detail on conditions provided below). Participants lived at the Appleton Institute Sleep Laboratory for 7 days (1 Adaptation Day, 5 Experimental Days and 1 Recovery Day). This study was approved by the Central Queensland University Human Research Ethics Committee (0000021914) and registered with the Australian New Zealand Clinical Trials Registry (12619001516178). Data from this study form part of a larger study, and the complete study protocol is published elsewhere [37].

### 2.2. Participants

Healthy adult males (*n* = 43) and females (*n* = 41) were recruited from the Adelaide region (Australia). Inclusion criteria were: being aged between 18-35 years, having a body mass index (BMI) range between 18-30 kg/m^2^, being a non-smoker, not currently being a shift-worker and not being diagnosed with a sleep disorder. Inclusion criteria were based on previous studies investigating cognitive impacts of breaking up sitting [1,2,37,38,39,40,41] to control for age and health-related factors which may have influenced outcomes. Participants were screened using a number of standard sleep, general health and physical activity questionnaires, with further information available here [37]. Participants were randomly assigned to one of four experimental conditions: (a) sitting and 9-h sleep opportunity, (b) sitting and 5-h sleep opportunity, (c) breaking up sitting and 9-h sleep opportunity or (d) breaking up sitting and 5-h sleep opportunity (See Table 1). During the experimental days (09:00-h–17:30-h), participants allocated to breaking up sitting conditions walked on a treadmill for 3 min every 30 min at a walking speed of 3.2 km/h. Participants allocated to sitting conditions remained seated during this time. All participants were provided with a 9-h sleep opportunity on the first night (Adaptation) and last night (Recovery) (22:00-h to 07:00-h). On the experimental nights, participants were assigned to one of two sleep opportunities: 9-h sleep opportunity (22:00-h–07:00-h) or 5-h sleep opportunity (02:00-h–07:00-h). Participants provided written informed consent and were compensated financially for their time at the conclusion of the study (AUD$780).

### 2.3. Experimental Procedure

The participants lived in a sound-attenuated and temperature-controlled (21 ± 2 °C) sleep laboratory for seven consecutive days and nights. During the day, light levels were kept at >100 lux, reflecting typical daily indoor light levels. On the adaptation day, participants were familiarised with all relevant questionnaires and all components of the driving task battery. On experimental days (E1–E5), participants completed a simulated work shift between 09:00 h and 17:30 h. Prior to the start of the simulated work shift (08:10 h) and immediately following the work shift (17:30 h), all participants completed a simulated commute, consisting of a 20-min driving task (Figure 1). A thigh-worn accelerometer (product details described in the next section) was attached to each participant on adaptation day, and it was removed on the recovery day. Physical activity levels were monitored continuously for the 7-day period via triaxial accelerometry. Only accelerometry data collected during each 20-min driving simulator task, as indicated in Figure 1, were used as input data for the classification by the CNNs. 

### 2.4. Measures

#### 2.4.1. Accelerometry

Leg movement data during each 20-min simulated drive were captured using triaxial accelerometry. The ActivPAL Micro 4 monitor (PAL Technologies, Glasgow, UK) was used, which is approximately 24 × 43 × 5 mm in size and weighs 9 g, with a sampling range of 20 Hz (+4 g to −4 g). The device was housed in a nitrile sleeve and attached to the anterior midline on the right thigh of each participant via an adhesive dressing and worn continuously throughout the 7-day study protocol. Raw accelerometer signals (x, y, z) were collected and processed from the raw uncompressed data files available through the ActivPAL software. The raw data signal files were extracted as CSV files for pre-processing, and training of the CNNs. Triaxial accelerometry was also used to objectively measure participants’ activity during the protocol, to confirm the ground truthing labels of the sitting or breaking up sitting conditions, using the ActivPAL algorithm to measure step count.

#### 2.4.2. Driving Simulator

Driving performance was simulated using the York Driving Simulator at the start and end of each participant’s work shift (York Computer Technologies, Kingston, ON, Canada). The driving simulator was presented on a single computer screen displaying a forward-facing view from the driver’s position. The controls of the driving simulator consisted of a brake and accelerator pedal on the floor below the computer and a steering wheel attached to the desk. Participants completed a 20-min driving task consisting of a simulated daytime rural drive with a speed limit of 110 km/h. The driving scenario presented in the simulation consisted of a single carriageway and a two-laned road with traffic travelling in both directions. Participants were instructed to use the right leg (the leg with the ActivPAL attached) to operate the brake and accelerator whilst completing the driving task. This driving simulator task has been used previously in driving and in sleep restriction studies [1,42,43].

#### 2.4.3. Sleep Monitoring

An activity monitor was continuously worn to objectively verify the sleep obtained in each condition (9-h or 5-h sleep opportunities). The Actical device (MiniMitter/Respironics, Bend, Oregon) is a wrist-worn activity monitor and is validated as a measure for sleep time and wakefulness [44]. The Actical device dimensions are 28 × 27 × 10 mm, and its weight is 17 g. This device uses a piezoelectric omnidirectional accelerometer with a range of 0.5 to 30 Hz to capture the frequency, intensity and duration of human movement [45]. The monitors were worn on the non-dominant wrist for the duration of the protocol (24 h per day)). Data were downloaded using Actical software (Phillips Actical MiniMitter, Respironics, Bend, OR, USA).

### 2.5. Statistical Analysis

Statistical analyses of sitting and sleep history (ground truthing labels) across experimental days (E1–E5) were performed using SPSS 26.0 Software (IBM Corp., Armonk, NY, USA). To confirm ground truthing labels for sleep history (i.e., distinct classes of 9-h and 5-h), an analysis of the total sleep time collected from actigraphy was conducted using independent *t*-tests. To confirm the ground truthing labels for sitting history (i.e., distinct classes of sitting or breaking up sitting), an independent t-test was performed for the step count retrieved from the ActivPAL. Data are reported as the Mean ± Standard Deviation. Statistical significance was set at *p* < 0.05. 

### 2.6. Accelerometry Data Preparation

Uncompressed acceleration data, sampled at a rate of 20 Hz from the thigh-worn ActivPAL 4 was downloaded using ActivPAL software (V8.10.8.76, PAL Technologies, Glasgow, UK) and labelled according to the time-stamped driving simulator files (two drives per day) using a customised script developed using Python (Version 3.4, Python Software Foundation). The raw accelerometer files contained data for the entire protocol. To isolate the relevant data, the drive start times contained in the driving simulator data files were cross-referenced to the time column in the raw accelerometer file. For each participant, there were 10 raw accelerometry files (representing each of the 20-min drives across the 5 experimental days). For the classification of prior sitting history (sitting or breaking up sitting), the first drive on E1 was excluded from the input data, as the experimental manipulation of sitting and breaking up sitting had not begun prior to this drive. For the classification of sleep history, the first drive on E1 was included in the input data, as the participants had a 9-h or 5-h sleep opportunity the night prior. Acceleration data from each axis was scaled to the range –4 to 4, with a value of 1 being equal to the acceleration due to gravity on Earth (~9.81 m·s^−2^), i.e., 1 g.

### 2.7. Neural Network Architecture

#### 2.7.1. DixonNet

Two network models were trained on the processed data. The first was an architecture initially presented by Dixon et al. [30] referred to as DixonNet hereafter (Figure 2). We chose to use DixonNet in this work due to the network’s efficacy as a classifier of an accelerometry dataset. The architecture of DixonNet is presented in Figure 1. The network operates on one-dimensional data consisting at each position of three channels: one channel for each of the x, y, and z axes found in the recorded accelerometry. The first two layers are convolutional, with each having 100 learned filters of a kernel size of 16 and a ReLU nonlinearity. These are followed by a max-pooling layer with a kernel size of 4, effectively quartering the length of the time-series provided while retaining the most significantly-activated features. Two more convolutional layers, configured identically to the first, conclude the spatial feature extraction portion of the network. Global average pooling, a dropout layer (*p* = 0.5) and one fully connected layer mapped these features to the number of outputs required for classification. In the current work, classification was binary; hence, the output size was set to one.

#### 2.7.2. ResNet-18

A second architecture was explored: a custom implementation of ResNet-18 [43]. This implementation was identical to that presented in He and colleagues’ work [46], except with each two-dimensional convolution layer being replaced with a one-dimensional equivalent. This modification was made to permit training on time series data (such as accelerometry data), which is one-dimensional, rather than two-dimensional, like image data. The architecture of ResNet-18 is shown in Figure 3. ResNet-18 has an initial convolutional layer with a kernel size of 7 × 7 and 64 filters [35]. It is followed by a batch normalization and ReLU activation step. These initial layers are then fed to a max-pooling layer with a kernel size of 3 and a stride of 2. The network then contains 4 residual blocks. Each block consists of two convolutional layers with a kernel size of 3 × 3 and 64, 128, 256 and 512 filters for blocks 1, 2, 3 and 4, respectively. These blocks are connected with the residual connections detailed in He et al. [46], providing a residual learning function. The final residual block is followed by a global average pooling layer (average pooling with output size of 1) and a fully connected linear layer. As with DixonNet, the architecture was used for binary classification and, hence, had an output size of 1. 

#### 2.7.3. Model Settings and Training

During training, rather than providing the whole drive task duration of data to the network at once, shorter windows of the data were taken from each drive sequence. This approach was chosen to decrease the drive time needed to achieve classification. For a given window length, any valid subset of consecutive data points was available for use as a labelled training sample, and thus, overlapping windows were present in the training data. Window length was therefore a configurable hyperparameter of the training process. However, window length also influences the practicability of the model, since a window length dictates the duration of data required to obtain an accurate classification after training. We used a window length of 4096 samples or approximately 200 s. A binary cross-entropy loss function was used to evaluate the networks’ learnable parameters on a batch-by-batch basis. The batch size was set to 64 windows. A variant of gradient descent called Adaptive Moment Estimation (Adam) was used to update the networks’ learnable parameters once per batch in the direction that minimised the loss function. The learning rate was initially set to 10^−4^ and reduced by a factor of 10 each time the loss on the training data failed to improve for 10 consecutive epochs. Data were randomly shuffled, and an 80/20 splitting procedure was performed using 5-fold cross-validation. The accuracy and F-score (the harmonic mean of recall and precision) were used to assess model performance on the validation data during training. Both models were developed in Python using PyTorch [47], and trained on a consumer-grade computer (2× GPU: Nvidia GTX 1080, CUDA Cores 2560, VRAM 8 GB).

### 2.8. Class Activation Mapping (CAM)

Class Activation Mapping (CAM) is a method of indicating which areas within a window of input data contribute the most to the CNNs’ resultant classification. In essence, CAM may be used to generate a heatmap across a window of data, highlighting key characteristic regions. In line with previous studies which have utilised CAM for both time-series and object recognition, a global average pooling layer was used penultimately in both networks explored, followed by a single linear fully-connected layer [35,36,48]. In this network, let $S_{k}\left(x\right)$ represent the output of the last convolutional layer on the channel *k*. The output of channel *k* in the global average pooling layer can be expressed as $f_{k}=\underset{x}{\sum }S_{k}\left(x\right).$ The input of the final SoftMax function can be defined as
(1)$$g_{c}=\underset{k}{\sum }w_{k}^{c}\underset{x}{\sum }S_{k}\left(x\right)$$
where $w_{k}^{c}$ is the weight in the fully connected layer representing the contribution of channel *k* to class *c*.

The class activation mapping for class c is represented as $M_{c}$.
(2)$$M_{c}=\underset{k}{\sum }w_{k}^{c}S_{k}\left(x\right)$$

A visual interpretive analysis of the CAM was conducted to identify repeated patterns within the signals associated with individual classes. The most confident samples were assessed and selected for class activation mapping using logits. All samples were ranked by the magnitude of raw network output before the activation function was applied. The five highest magnitude samples and the five lowest magnitude samples of each network and class were chosen for visual interpretation.

## 3. Results

### 3.1. Ground Truth Verification

Total sleep time across all experimental days was different (*p* < 0.000) between the 9-h sleep history class (08:00 ± 00:40 h) and the 5-h sleep history class (04:34 ± 00:18 h). Step count across all experimental days was different (*p* = 0.031) between the sitting class (1231 ± 661) and the breaking up sitting class (7016 ± 742).

### 3.2. Model Performance

Table 2 presents a summary of the performance of each model for the classification of sitting history and sleep history. Figure 4 presents the average summary of each fold for model performance measured via F-score and training loss across time (epoch). The confusion matrices for each model are presented in Figure 5. Overall, ResNet-18 out-performed Dixon Net in terms of classification accuracy and for model performance, as seen in Figure 4.

### 3.3. Class Activation Mapping (CAM)

The class activation mapping for sitting and sleep history for both models is shown in Figure 6. Two example signals are provided in Figure 6 from each model and class combination representing exemplars (in the confidence of prediction) of each class prediction. The CAM for DixonNet classifying the sitting history shows that a key feature of the signal may be postural changes, as seen by the changes in acceleration (g) along the y and z signals. Postural changes are visible through differences in the way that the background acceleration due to gravity (1 g force) is distributed over the three axes. Different distributions of acceleration caused by the earth’s gravity therefore show different passive thigh angles, indicative of postural changes [49]. In contrast, ResNet CAM for activity highlights more movement along all three directions of the accelerometer. The CAM of the 5-h sleep history class classified by DixonNet displays periods of inactivity punctuated by sudden and rapid movements. In comparison, the CAM of the 9-h sleep history class for DixonNet shows inconsistent and smaller movements across time. The CAM of the ResNet model for the 5-h sleep history class highlights frequent rapid movements across time. The 9-h sleep history CAM, as classified by ResNet-18, demonstrates smaller movements across time, similar to the 9-h sleep history CAM from DixonNet.

## 4. Discussion

This study investigated the feasibility of applying a deep learning approach to classify prior sitting and sleep history from raw accelerometry data captured during a simulated drive. Two CNNs, DixonNet and ResNet-18, were able to accurately classify sitting and sleep history from raw accelerometry data without prior filtering or signal processing. These results demonstrate the potential application of raw accelerometry data to classify behaviours (i.e., sitting and sleep history) before they can become problematic for driving performance, with the potential to extend this research to future real-time classification.

In this study, ResNet-18 outperformed DixonNet for the classification of sitting and sleep history. ResNet-18 was more accurate than DixonNet and demonstrated less variability across the evaluation metrics. Both models were able to learn from the raw accelerometry data whilst avoiding overfitting, often seen with unbalanced datasets or due to overtraining, and which is a common problem for many deep learning approaches [50,51]. The high accuracy across the five-folds and the consistently decreasing loss shown in the current study demonstrates that this was robust and generalised learning. The model performance outcomes presented in this study are in line with previous CNN research classifying accelerometry data (e.g., fall detection, surface detection), which achieved high accuracy rates (95–97%) whilst demonstrating generalised learning [30,35]. The robust learning shown by both CNN models in this study further speaks to the suitability of applying a deep learning approach to sleep and activity classification using accelerometry data. 

The class activation mapping (CAM) presented in this study visualizes key regions of the classified accelerometery signals that contributed strongly to the final classification. This visualization method provides a means to interpret the difference in physical movement seen between activity classes (i.e., sitting or breaking up sitting) and sleep history classes (i.e., 9-h or 5-h) during drives. The current study builds on previous work utilising CAMs for the visualisation and analysis of accelerometry data, which have, up until now, primarily focused on fall detection and classification [35,48]. The current study extends the application of CAMs to visualize the efficacy of a deep learning approach to the application of classifying driver states (e.g., inactive, sleepy). The CAM approach presented in this study provides a new way to visualize the previously researched physiological and behavioural impacts of sleep restriction (e.g., slower reaction time) on drivers [4,8,23,52,53]. The rapid and inconsistent movements displayed in the 5-h sleep history class by both networks may be indicative of the slower reaction times experienced by drivers who are sleep restricted [8,15,54,55]. In contrast, the consistent and smaller movements displayed in the CAM for the 9-h sleep history class from both models may indicate increased psychomotor functioning with more controlled inputs while driving (e.g., smoother braking and accelerating patterns) [15,54]. 

The CAM for the sitting history emphasises that different features were activated more strongly by the two models. The DixonNet CAM displays periods of limited or no movement for sitting and breaking up sitting classes, which may indicate that this model recognised and learnt postural changes rather than movement changes. In comparison, ResNet-18 was more highly activated by movements rather than postural changes. The movement patterns identified by ResNet-18 may reflect the differences in psychomotor function (e.g., reaction time) reported after breaking up sitting in previous research [11,56,57,58]. Previous studies have reported improvements in psychomotor function, reaction time, and attention for cognitive tasks after breaking up sitting with light-to-moderate intensity physical [13,57,59]. The highlighted patterns in movement via the CAM of ResNet-18 might be reflective of the improvements in psychomotor function. Unlike sleep history classification, there is limited literature outlining the exact characteristics of physical movements associated with potential behavioural changes during driving as a result of activity levels (i.e., sitting or breaking up sitting) during the day. This is the first study to report CAM analyses on physical leg movements, and it shows clear differences associated with activity levels throughout the day. 

The deep learning approach in this study presents accuracy rates (75.7–88.6%) in line with previous deep learning physiological fatigue detection studies, which have achieved accuracy ranges between 74.9–96.0% [19,60]. Other studies utilising a deep learning approach for fatigue detection have developed customised hardware (e.g., Arduino smartwatch device, wearable glove device), require complex post-processing of data and recording multiple input signals (electroencephalogram, photoplethysmography) [27,61,62]. The methodology used in this study offers an alternative to customised hardware solutions and multi-signal approaches for fatigue detection whilst still maintaining classification accuracy rates in line with or higher than previous research. Additionally, the approach presented here may be more applicable to real-time classification of driver states than the previously mentioned classification methodologies, which are computationally less efficient and unsuitable for real-time classification. 

The placement of the accelerometer is an important aspect to consider in physical activity and movement analysis. In previous studies, accelerometers have been positioned on the hip area, as placement closest to the centre of mass generally better reflects movement of the entire body, aiding classification [63]. However, centralised locations may not be useful in classifying subtle changes in movement, such as those involved in driving (e.g., at the extremities and hands/feet, which are used for driving) [63]. In the current study, thigh-worn accelerometer devices were chosen, as they are particularly useful for classifying driving behaviour for braking and accelerating events whist also capturing postural information. However, other placements, such as at the wrist, may provide information about physiological responses from steering [62,64]. Additional or new patterns of movement to detect behaviours (i.e., prior sitting or sleep history), could benefit from alternate placements of accelerometers in future classification studies.

There are limitations to this study that should be considered when interpreting the results. The sampling rate of the accelerometer used in this study was 20 Hz, a lower sampling rate than previous accelerometer classification studies (between 30–1024 Hz) [30,63,65]. Accurate assessment of movements requires the sampling signal to be double that of the signal of interest and for acceleration signals related to movement of the body to occur at or below 10 Hz [63]. Therefore, while the sampling rate used in this study was within the required range, a higher sampling rate may have improved the accuracy of the results of the models due to an increase in available data, in line with similar accelerometer classification studies [30,65,66]. This study was conducted with young healthy participants, limiting generalisability to real-world sedentary drivers, including older adults and people with health issues (e.g., cardiometabolic disorders, diagnosed sleep disorders) [67,68,69]. The monotonous driving scenario, consisting of low traffic density and high speeds used in this study, was chosen as it provides a sensitive measure of sustained attention and fatigue, particularly under sleep restriction [70]. However, the driving scenario used in this study may not be representative of a wide range of driving situations, particularly in cities with high-density traffic scenarios requiring constant braking and accelerating [71,72]. Therefore, the patterns of movement learnt by the models during this driving scenario may not be as applicable or accurate if applied to data collected in different traffic conditions.

There are several recommendations to build on the current findings. Adapting the current approach to detect prior sleep of a driver within the shortest time frame (i.e., smallest window size) would be a logical next step in this research. Detecting and classifying the sleep history of a driver could potentially highlight impairments sooner, before they put the driver or other road users at risk [62]. While the input of smaller window sizes may allow for earlier detection and classification of sleep history, this can lead to challenges such as reduced accuracy due to insufficient data input. Additionally, extending this research by training models to classify other sleep durations (i.e., 2 h per night) would be of benefit to identify fatigued drivers in populations such as professional drivers or new parents, where sleep opportunities may be less than 5 h per night [6,73]. 

Additional considerations to extend this research would be understanding the complexities of applying a CNN classification approach to real-world scenarios, rather than in a highly controlled laboratory environment. Although accelerometers are widely available in many smart devices and could be applied to classifying behaviours of drivers in uncontrolled environments (e.g., real-world driving scenarios), this also carries challenges of how to account for variables which may hinder the classification of driver states (i.e., sedentary or sleepy). Driving in a moving car creates additional noise in the accelerometer signal due to the vibration and inertial forces of the vehicle’s movements (i.e., speed variability of the vehicle) [74]. Both speed variability and vibration are unpredictable forces to control for in accelerometry data alone, as these factors would vary considerably based on road conditions and vehicle type [74]. Speed variability is particularly prevalent on urban roads due to rapidly changing traffic conditions, such as changes in traffic density, which requires constant and rapid changes in vehicle speed [75]. However, secondary signals, such as a global navigation satellite system, have been used in conjunction with accelerometers to account for speed variability in studies focusing on road surface classification [76,77]. Although there are a number of considerations to build on this approach to make it applicable to real-world driving scenarios, this research demonstrates the potential to classify changes in the driver’s state in real-time from thigh movements.

## 5. Conclusions

This study has demonstrated the effectiveness of raw accelerometry data for the classification of prior sitting and sleep history. Accuracy scores of up to 88% for the classification of prior sitting and sleep history from raw accelerometery were achieved without the need for extensive filtering or hand-crafted feature extraction. The findings further extend how accelerometry can assess impairments to safety critical tasks such as driving, providing a potential alternative to current driver fatigue detection methods using raw accelerometry data. Further, the results suggest that prior sitting history may lead to changes in movement patterns during driving, which are detectable with a deep learning approach. Whilst previous deep learning approaches such as computer vision may detect the appearance of driver states (e.g., fatigue, drowsiness), the approach presented here may provide more insight into the physical impacts of sitting and sleep during driving. Further, this information could be used to monitor and detect early signs of psychomotor impairment, allowing drivers to be better informed about their ability to safely operate a vehicle.

## Figures and Tables

**Figure 1 sensors-22-06598-f001:**
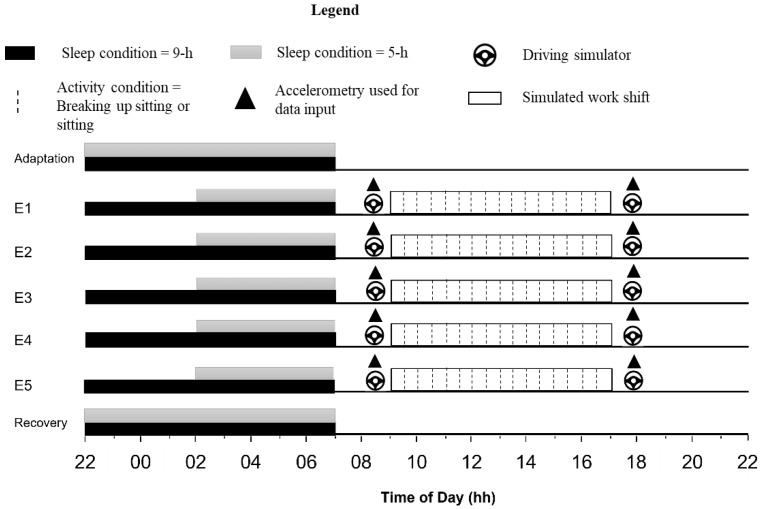
Experimental design for the 7-night protocol.

**Figure 2 sensors-22-06598-f002:**
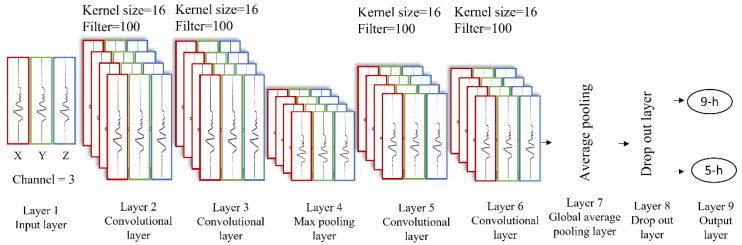
DixonNet convolutional neural network architecture. The final fully connected layer acts as the output layer for binary classification (i.e., sitting or breaking up sitting or 9-h or 5-h sleep history).

**Figure 3 sensors-22-06598-f003:**
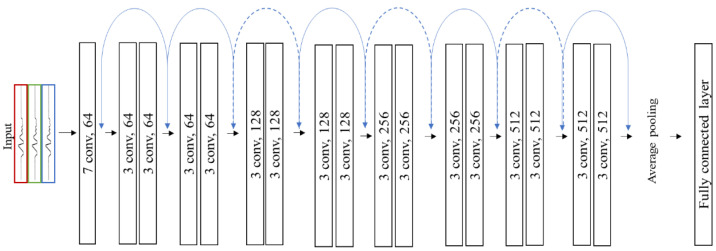
ResNet-18 convolutional neural network architecture. The dotted blue lines represent the residual connections (or skip connections) utilised in this network. Conv = convolutional layers. The final fully connected layer acts as the output layer for binary classification (i.e., sitting or breaking up sitting or 9-h or 5-h sleep history).

**Figure 4 sensors-22-06598-f004:**
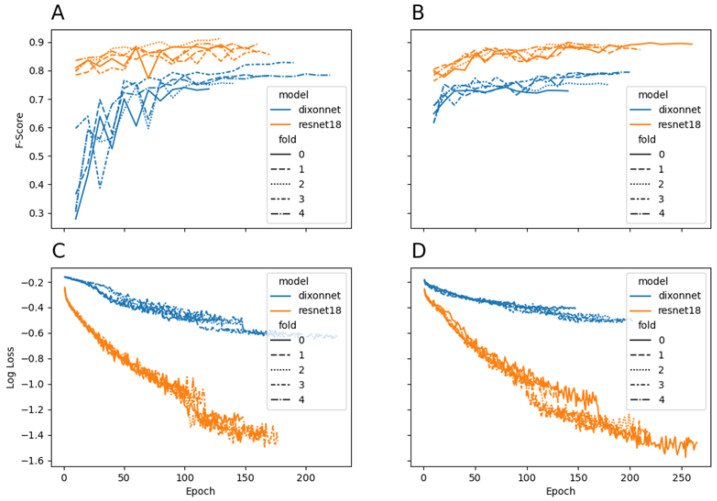
The average model performance across each fold for sitting and sleep history classification. Panel (**A**) evaluation accuracy (F-score) for sitting history; Panel (**B**) evaluation accuracy (F-score) for sleep history; Panel (**C**) natural log of the training loss for sitting history; Panel (**D**) natural log of the training loss for sleep history.

**Figure 5 sensors-22-06598-f005:**
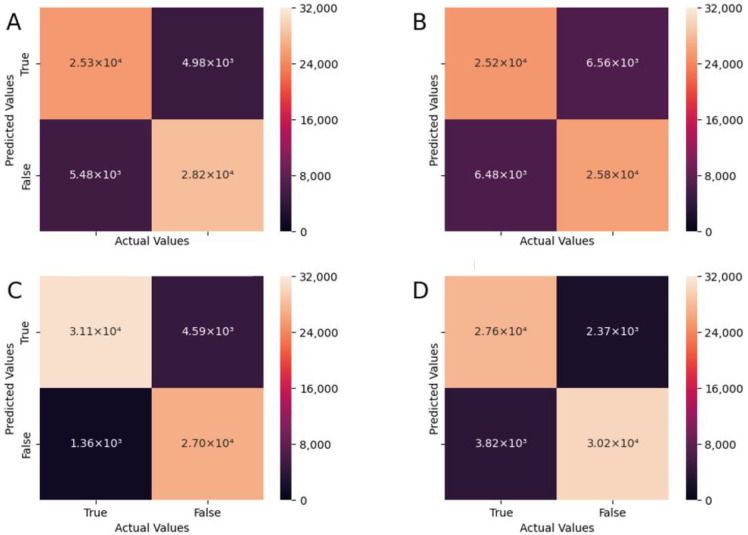
Confusion matrices for each model for sitting and sleep history classification. (**A**) DixonNet sitting history; (**B**) DixonNet sleep history; (**C**) ResNet-18 sitting history; (**D**) ResNet-18 sleep history.

**Figure 6 sensors-22-06598-f006:**
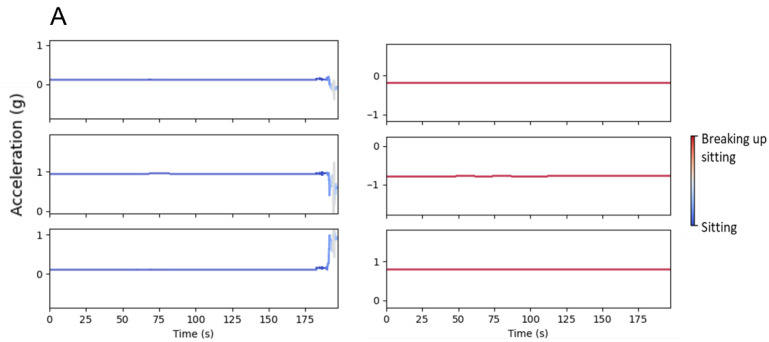
Class activation maps of classified acceleration signals produced from CNN outputs of DixonNet and ResNet-18 for sitting and sleep history. (**A**) DixonNet sitting history; (**B**) ResNet-18 sitting history; (**C**) DixonNet sleep history; (**D**) ResNet-18 sleep history. Each individual plotted signal represents the x, y and z axis of the accelerometer, respectively (from top to bottom). The colour of the line is representative of the strength in confidence of the class prediction (sitting or sleep history). The Y axis is centred on the mean of each signal ±0.5 g, meaning there is a constant 1-g scale for each plot. The heat map bar represents the strength of prediction of each class.

**Table 1 sensors-22-06598-t001:** Number of participants in each experimental condition (class).

		Sleep History	
		9-h	5-h
Sitting history	Sitting	22	22
	Breaking up sitting	20	20

**Table 2 sensors-22-06598-t002:** Summary performance of each model (across the 5-folds) for each class. Results are presented as the Mean ± SD.

Model	Class	Accuracy (%)	F-Score
**Dixon Net**	Sitting history	77.24 (±2.61)	0.76 (±0.03)
	Sleep history	75.71 (±2.69)	0.76 (±0.02)
**ResNet-18**	Sitting history	88.63 (±1.36)	0.88 (±0.01)
	Sleep history	88.63 (±1.15)	0.88 (±0.01)

## Data Availability

Data are available from the authors upon request. The code is available here: https://github.com/keeeal/sleepy-and-sitting.

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
