# Peer review of "A Deep Learning Approach to Classify Sitting and Sleep History from Raw Accelerometry Data during Simulated Driving"

_sensors, 2022, doi:10.3390/s22176598_

Round 1

Reviewer 1 Report

Here some suggestions to improve the current work:

- For more visibility, the authors would present a figure of workflow of the proposed approach also for the conducted expriment which can help readers to more understand the analysis.

- Replace 'machine learning' when you mean 'Deep learning'

- I have a question, in Table 2 page 8, the authors obtain the same value for both classes 0.77 for DixonNet and 0.88 for ResNet-18. 

Can the authors provide the confusion matrix.

Reviewer 2 Report

The content of this manuscript is quite interesting and attractive. The literature review is complete, the description of research methodology and discussion is in-detailed. Just few points should be revised/improved before the acceptance for publication.

1. The resolution of Fig.1 should be improved.

2. The resolution of Fig.4 is poor, mush be improved.

3. The description of Fig.4 is lengthy, should be concise. Some illustration could be moved to the main text of content.

4. In conclusion, some important findings and some key quantitative results should be added.

Reviewer 3 Report

The topic undertaken by the authors is very interesting and important from many points of view. The article is written correctly, although it contains some errors. Therefore, in order to improve the scientific quality of the article, some corrections should be made, as listed below:

1. There is no "Literature Review" section in the article - some things have been included in the "Introduction" section, but in my opinion it should be separated.

2. There is no explanation why the authors adopted such inclusion criteria? Why was the age of the participants between 18-35 assumed? And not, for example, 18-30?

3. Is Figure 1 authored and does it concern own research? It is a bit unreadable.

4. Formulas, eg those contained in subsection 2.8, should be numbered to make them easier to refer to.

5. The results in table 2 and in figure 3 are in no way described and explained in subsection 3.2.

6. The title of figure 3 should be short, and the explanations of the figure should be written below or even in the text, and should not be part of the title. The same goes for figure 4.

7. Were there any limitations to the research? Can the obtained results be generalized? - Conclusions should be a bit more elaborate.

Minor linguistic, punctuation and stylistic errors were noticed in the text of the manuscript - when improving the article, please pay attention to it and correct it accordingly.

Round 2

Reviewer 1 Report

Thanks for your responses!!